# Walking Turn Prediction from Upper Body Kinematics: A Systematic Review with Implications for Human-Robot Interaction

**Antonio M. López \* , Juan C. Alvarez and Diego Álvarez**

Multisensor Systems and Robotics Group (SiMuR), Department of Electrical, Electronic, Computer and Systems Engineering, University of Oviedo, C/ Pedro Puig Adam, 33203 Gijón, Spain; juan@uniovi.es (J.C.A.); dalvarez@uniovi.es (D.Á.)

\* Correspondence: amlopez@uniovi.es; Tel.: +34-98-518-1994

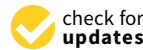

**Featured Application: Upper body kinematics (Head Yaw and Trunk Roll from evidence so far) can be used to anticipate walking turns in the short-term, and can therefore be used to improve the interaction between humans and robots in close proximity.**

**Abstract:** Prediction of walking turns allows to improve human factors such as comfort and perceived safety in human-robot interaction. The current state-of-the-art suggests that upper body kinematics can be used for that purpose and contains evidence about the reliability and the quantitative anticipation that can be expected from different variables. However, the experimental methodology has not been consistent throughout the different works and the related data has not always been given in an explicit form, with different studies containing partial, complementary or even contradictory results. In this paper, with the purpose of providing a uniform view of the topic that can trigger new developments in the field, we performed a systematic review of the relevant literature addressing three main questions: (i) Which upper body kinematic variables permit to anticipate a walking turn? (ii) How long in advance can we anticipate the turn from them? (iii) What is the expected contribution of walking turn prediction systems from upper body kinematics for human-robot interaction? We have found that head yaw was the most reliable kinematical variable from the upper body to predict walking turns about 200ms. Trunk roll anticipates walking turns by a similar amount of time, but with less reliability. Both approaches may benefit human-robot interaction in close proximity, helping the robot to exhibit appropriate proxemic behavior interacting at intimate, personal or social distances. From the point of view of safety, they have to be considered with caution. Trunk yaw is not valid to anticipate turns. Gaze Yaw seems to be the earliest predictor, although existing evidence is still inconclusive.

**Keywords:** human-robot interaction; movement prediction; pedestrian navigation; body kinematics

---

## 1. Introduction

Human motion prediction is essential for the safe and effective interaction between humans and robots in collaborative spaces. For instance, it has been outlined as a fundamental need for robots that consciously navigate in the presence of humans, in order to develop a "human-aware" navigation strategy [1]. Additionally, fueled by recently proposed specifications (ISO/TS 15066:2016—Robots and robotic devices—Collaborative robots), it will likely be essential in the near future for the realization of the "intimate collaboration" or the "close proximity interaction" paradigms described by [2,3], respectively. In particular, studies have shown recently that human motion prediction technologies have the potential to improve the efficiency of the team and the perception of

safety and comfort by the user in different settings in which persons and robots share the same physical workspace [4,5]. Likewise, human motion prediction may also be relevant in assistive devices (active prosthesis/orthosis/exoskeletons) to switch between different locomotion modes [6,7], by changing the control algorithms or by modifying the mechanical properties of the assistive devices [8].

Automatic anticipated detection of walking turns from body kinematics is a plausible approach as it is a well-known fact in neurosciences that postural adjustments are initiated prior to the actual change in the heading direction, and that these adjustments are reflected in the kinematics of body segments (see for instance [9]). It is also a feasible approach as nowadays there are different reliable technologies that allow to sample these variables in real time, with optical sensors and Inertial Measurement Units being the most representative. Some pioneering applications have confirmed the validity of this approach [10,11]. Finally, it is a promising approach for human robot collaboration as it may be used to schedule and execute robotic navigation plans that are compliant with the expected trajectories of the user [1], furthermore contributing to improve the safety and comfort of the interacting humans [11].

In particular, the state-of-the-art comprises a vast amount of knowledge about the role of upper body kinematics during locomotion steering that suggests that turn detection systems can be implemented from them, especially from gaze, trunk and head kinematics (see Figure 1). For instance, it has been shown that head anticipates motion direction to provide a stable reference frame that helps to coordinate the motion of the other body segments [12]. Likewise, it was argued that trunk roll and yaw control help the lower segments to further redirect the Body Center of Mass in the new direction of travel [13]. Moreover, head yaw and trunk tilt were shown to be connected to eye movements in order to stabilize the optic flow needed to control the upcoming steering maneuver [14]. Different steering maneuvers, such as lane-changes [15] or obstacle circumventing [16], were shown to be eventually preceded by some postural adaptations in the upper trunk. Steering under special circumstances, such as manual wheelchair navigation [17] or while pushing a cart [18], changed the preceding upper trunk motion as compared to normal biped walking.

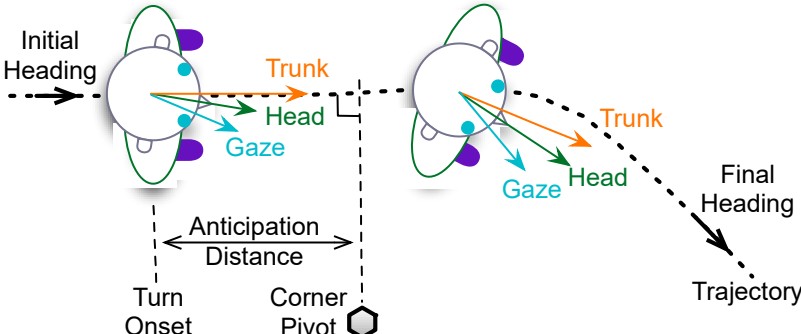

**Figure 1.** At the beginning and at the end of the turn all the upper body segments are aligned with the actual trajectory of the subject (dotted line). However, during the preparation and execution of the turn, certain postural adjustments are made sequentially which are reflected in the kinematics of different variables of the upper body (gaze, head, trunk).

However, it is difficult to interpret this knowledge from the point of view of turn prediction systems because the original studies were fundamentally oriented to study human motor and cognitive controls. Additionally, the experimental methodology was not consistent throughout the different works and the relevant data was spread over different scientific sources in an unstructured way and not always given in an explicit form, with different works containing partial, complementary or even contradictory results.

Thus, it is not clear yet what upper body kinematical variables can be reliably used to predict a change of direction of a person during normal walking, the extent of anticipation that can be reached

from them or the potential contributions that anticipated turn detection from upper body kinematics could present on human-robot interaction.

To provide an answer, in this paper we collected, structured and reinterpreted the relevant information comprised in the state-of-the-art about human performance during walking turns in order to answer three main questions:

- Which kinematic variables (upper body) reliably permit to anticipate a turn?
- How long in advance can we anticipate the turn from them?
- What is the expected contribution of upper body kinematics –based turn prediction systems in human-robot interaction?

## 2. Materials and Methods

### 2.1. Paper Collection

IEEE Xplore and SCOPUS were selected as the reference database in order to have a broad coverage that includes both scientific and technological fields. During a preliminary exploration, we found two main topics containing information of interest for our purposes. The first was concerned specifically with anticipatory postural adjustments to turning. The other analyzed human kinematics during turning and sometimes included information about pre-turn maneuvers that can be interesting to analyze anticipatory adjustments. Therefore, the final search was based on queries from the following groups of keywords: (GAIT and ANTICIPATION and TURNING); ((GAZE or HEAD or TRUNK or HIP) and TURNING and KINEMATICS). Additionally, we defined different synonyms for the search, such as WALK in place of GAIT, or STEERING or CHANGE OF DIRECTION in place of turning.

The inclusion criterion for the screening of papers was as follows:

- The experiments included a change in direction during normal walking of healthy adult subjects. Sustained circle-walking was not considered.
- The results included quantitative values about the time/spatial anticipation from kinematics (positions, velocities or accelerations) of variables of upper body segments.
- The studies were published in English.
- The studies were published in a journal.

An initial search was conducted in January 2016 (SCOPUS, IEEE Xplore), which included 2015. 397 raw references. Title and abstract were independently reviewed by the three authors and decision about their relevance for the study was taken after consensus. 24 papers were selected. Seven of them were removed after a preliminary read.

During the subsequent execution of the study, the authors individually screened (title, abstract, preliminary read) new papers found using varied sources (SCOPUS, Web of Science, IEEE Xplore and Google Scholar). These new papers were considered at different literature updates (September 2017, February 2018). After a detailed analysis, one paper was incorporated into the study.

Selected papers were distributed among the authors. Each author collected the relevant data from the assigned articles, with the final supervision of the other two.

From the finally considered works, nine of them did not provide precise quantitative results and were not used for quantitative analysis. Relevant findings from these works were qualitatively considered in the discussion when appropriate.

Figure 2 summarizes the process using the PRISMA (Preferred Reporting Items for Systematic Reviews and Meta-Analyses) flow diagram.

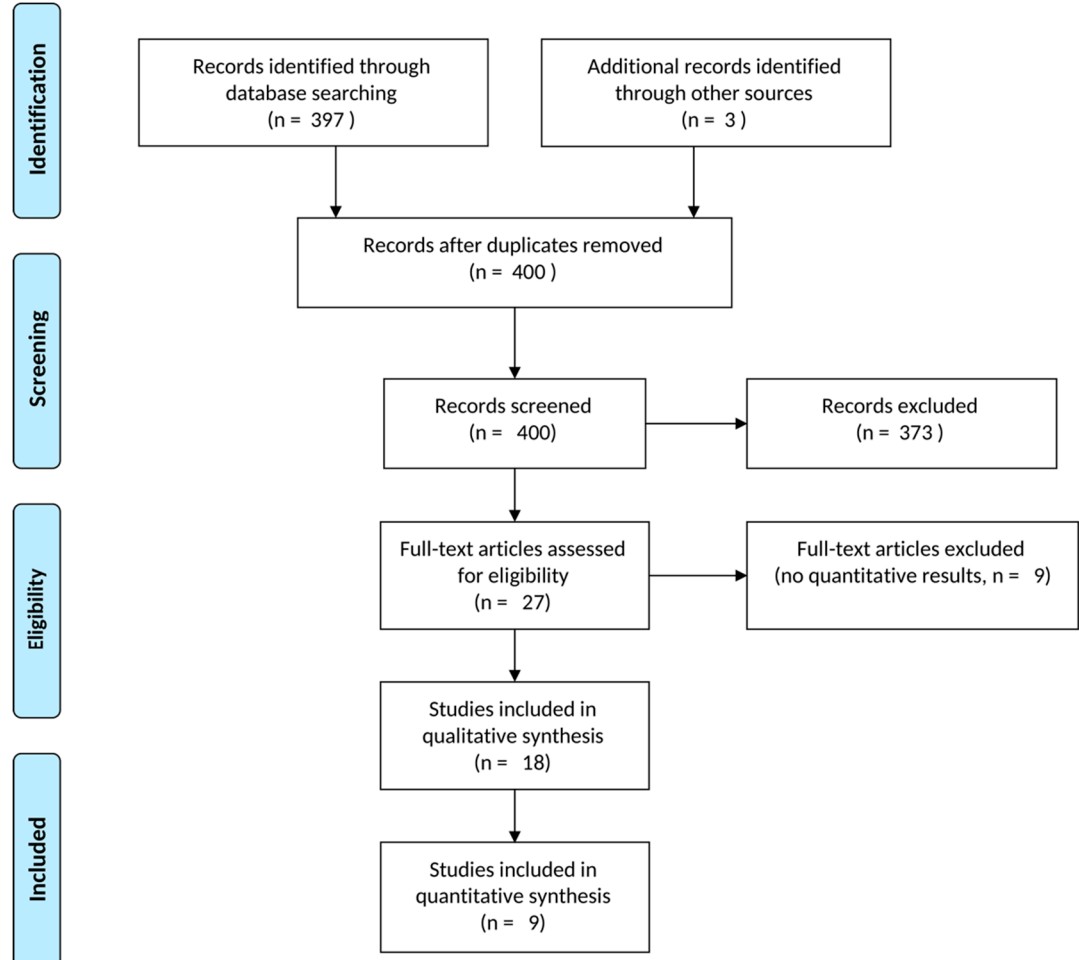

**Figure 2.** Flow Diagram of the study according to PRISMA methodology.

*2.2. Definition of Variables*

We defined the variables of interest to our study attending those included in the considered works. We have found a great deal of work about the estimation of Gaze Yaw (GY), Head Yaw (HY), Trunk Yaw (TY) and Trunk Roll (TR). Trajectory Heading (TJH) and Body Center Of Mass Medio-Lateral Displacement (MLC) were included in some references, and will be used in our study to compare the feasible anticipation with the previous variables.

Some other variables were included sparely in some studies. This is the case of Pelvis Yaw [19] or Gaze/Head Elevation [20]. However, experimental data about them was scarce and, moreover, existing studies reported no significant anticipation to the turn; thus, they were not included in our later analysis.

*2.3. Classification of Works*

After a thorough analysis of the contents of the different works, we found two main classes of studies that were used in our later analysis:

- Time synchronization. These studies addressed the synchronization of kinematical variables from upper body segments during turning maneuvers, eventually including the trajectory heading or the medio-lateral displacement of the body center of mass.
- Geographical anticipation. These studies addressed the anticipation of kinematical variables from upper body segments in reference to external physical world landmarks, usually a corner pivot located at the turning point.

## 2.4. Compilation of Experimental Conditions

Nearly all the studies used optical motion capture systems to sample body kinematics. Technical details are included in Table 1. The second column contains the addressed variables. Columns three to five describe the number and position of head/trunk/center-of-mass(COM) markers used in the optical motion capture system for the estimations. Column six describes the method used for the estimation of the actual trajectory of the subject. Column seven includes the technique used to quantify the anticipation between variables (see Figure 1). CC (Cross Correlation) was used for studies that used cross correlation to obtain the delay between the compared variables. TO was used for studies that compared the time the turn onset was detected in the compared variables. Experimental details are included in Table 2 (trials, people, age, walking speed and kind of turning). The first column of Tables 1 and 2 contains the reference of the work.

**Table 1.** Kinematical variables and estimation details in the studies based on optical motion capture systems. Gaze Yaw (GY), Head Yaw (HY), Trunk Yaw (TY), Trunk Roll (TR), Trajectory Heading (TJH) and Body Center Of Mass Medio-Lateral Displacement (MLC).

| Reference | Variables | Head Markers | Trunk Markers | COM Markers | Trajectory Estimation | Time Difference |
|---|---|---|---|---|---|---|
| [12] | HY, TJH | 2: Frontal and occipital (antenna) | | | Head midpoint displacement | CC |
| [21] | HY, TY, TJH | 2: Frontal and occipital (antenna) | 2: shoulders (antenna) | | Head midpoint displacement | CC |
| [22] | GY, HY, TJH | 2: Frontal and occipital (antenna) | | | Head midpoint displacement | CC |
| [23] | HY, TY, TR | 3: Eyes border and chin | 3: acromions + sternal notch | | | TO |
| [24] | HY, TY, TR, MLC | 3: Eyes border (antenna) and chin | 3: acromions + xiphoid process | 28: 14-segments anthropometric model | | TO |
| [25] | HY, TY, MLC | 3: Eyes border + chin | 3: acromions + xiphoid process | By using trunk markers | | TO |
| [26] | GY, HY | | | | | TO |
| [16] | HY, TY, TR, MLC | 3: not specified | 3: acromions + xiphoid process | 9: 3-segments (head, neck, trunk) model | | TO |
| [27] | HY, TJH (video) | 2: frontal and occipital (antenna) | | | Head midpoint displacement | TO |
| [28] | HY, TY, TR, MLC | 3: not specified | 3: sternal notch | By using trunk markers | | TO |
| [29] | GY, HY, TY | 4: back and front of temples | 2: acromio-clavicular joints | | Head midpoint displacement | CC |
| [19] | HY, TJH | 3: temples + glabella | 3: acromions + sternum | | Trunk markers midpoint | CC |
| [20] | GY, HY, TJH | 43: VICON® plugin gait model | 43: VICON® plugin gait model | | Isobarycenter of the pelvis displacement | CC |
| [30] | HY | 3: helmet | | | Proyection of the head | TO |
| [31] | HY, TY | 4: helmet | 4: backpack | | | TO |
| [17] | GY, HY, TY | 3: not specified | 3: sternal notch | | | TO |

An additional study was found where the orientation of some upper body variables (head, upper back and lower back) was estimated from Inertial Measurement Units (IMU) [10]. An IMU was placed in each of the addressed body segments. Ten subjects (mean age 30.9, standard deviation 4.3 years) performed 49 trials involving turns at $-90°$, $-45°$, $-22°$, $0°$, $22°$, $45°$, $90°$. Turn onsets were defined on those variables by orientation and velocity thresholds and used to analyze their anticipation to the actual turning time, estimated from the visual analysis of the experiments using motion capture systems. IMUs were also used in [32] to study the anticipation between head and hip orientation. Seven subjects (age 25–31) were asked to walk forward, then to continue straight or turning right/left (90°). Thirty trials were recorded for each subject, 10 of them in each turn condition. To determine turn onset times, pelvis and head yaw behavior were modeled as second-order linear time invariant systems with a nonlinear delay. Model parameters were identified from each experiment using widely accepted computational methods (MATLAB® Identification Toolbox). The value identified for the delay was used to time the turn onset from the start of the experiments.

**Table 2.** Experimental conditions used in the studies based on optical motion capture systems.

| Ref. | Trials $\times$ People/Age (mean $\pm$ std)/Speed (m/s) | Turning Angle ($°$) or Turn Type |
|---|---|---|
| [12] | $6 \times 5/28 \pm 2.5/(0.69 \pm 0.03; 1.01 \pm 0.03)$ | |
| [21] | $2 \times 6/-/-$ | 90 |
| [22] | $3 \times 6/33 \pm 4/1.15 \pm 0.15$ | 90 |
| [23] | $5 \times 6/22.5 \pm 2.1/-$ | 20, 40, 60 |
| [24] | $20 \times 5/24.8 \pm 4/-$ | $-60, -30, 30, 60$ |
| [25] | $20 \times 5/19.7 \pm 1.2/1.3 \pm 0.2$ | 30, 60 |
| [26] | $10 \times 7/24.8 \pm 4/-$ | $-60, -30, 30, 60$ |
| [16] | $10 \times 6/26.3 \pm 2.9/1.3 \pm 0.2$ | Obstacle |
| [27] | $1 \times 12/33 \pm 3/(0.8 \pm 0.07; 1.2 \pm 0.05, 1.6 \pm 0.06)$ | 90 |
| [28] | $30 \times 3/20.7 \pm 2.9/1.30 \pm 0.1$ | $-40, 40$ |
| [29] | $27 \times 10/29.9 \pm -/-$ | Limaçon, eight (normal and extended), cloverleaf, free-left and right |
| [19] | $13 \times 6/22$ to $33.5/1.08$ to $1.29$ | $-90, 90$ |
| [20] | $8 \times 10/30 \pm 5.2/$ | Eight, limaçon |
| [30] | $24 \times 10/32.1 \pm 11.9/-$ | |
| [31] | $8 \times 24/24 \pm 2.5/(1.4; 2)$ | 45, 180 |
| [17] | $2 \times 8/26.9 \pm 6.4/1.4$ | 90 |

## 2.5. Data Analysis

We decided to base our primary analysis on the studies where optical motion capture systems were used to sample body and trajectory kinematics (Tables 1 and 2).

In general, results were reported in numerical format in the original works using the mean and standard deviation of the values collected for the analyzed parameter across experimental trials. However, in some studies [23–25,28], results were reported using bar graphs showing the mean value and the corresponding standard error. We identified from them the corresponding numerical values by hand. To standardize the later analysis, we then estimated the standard deviation as the product of the reported standard error times the square root of the number of effective experiments (Trials $\times$ People, Table 2-Column 2). Additionally, we estimated the anticipation in distance to the corner pivot from the anticipation times reported in [17] using the average walking speed of subjects in the experiments (Table 2-Column2) .

In some studies, onsets of different variables were timed from a common external reference event. We have used that data to calculate the synchronization between the variables themselves. For that purpose, we have calculated (see Supplementary Materials) the mean delay between the variables by subtracting their mean delay from the reference event. $s_{1,2} = \sqrt[2]{s_1^2 + s_2^2 - 2\rho s_1 s_2}$ was used to estimate the standard deviation of the delay between variables $s_1$ and $s_2$, where $s_1$ and $s_2$ are the reported standard deviations of the delay of each variable from the common reference event and $\rho$ is the correlation factor between the variables. In our study we have assumed that kinematical changes in the upper body were highly positively correlated. Thus, we decided to use a correlation factor of $\rho = 0.85$ [33,34].

Anticipation values from different studies were aggregated when appropriate using a random effects model (R statistical analysis package, rma.uni() function from the metaphor package, default Restricted Maximum-Likelihood Estimator method, [35]; see Supplementary Materials). Mean anticipation between pairs of variables were analyzed. p-values and the 95% confidence interval reported by the software were used to find those pairs where the null hypothesis (mean anticipation equals zero) could be rejected.

## 2.6. Data Collection and Preprocessing

Table 3 contains the collected mean and standard deviation (column 2) of time delays of onsets on Gaze Yaw (GY), Head Yaw (HY), Trunk Roll (TR) and Medio-Lateral COM (MLC) displacement from a reference event in a preparatory step. Different reference events were used in the analyzed

studies. In [26], the contralateral toe-off of the step that preceded the turn was used for that purpose. Other studies were based on the right foot contact [23–25] or the heel contact that corresponded to the initiating turning step [28].

**Table 3.** Time delays (mean ± standard deviation) from a reference event in a preparatory step to turn onsets on different variables (GY, HY, TY, TR, MLC) collected from references specified in Column 4. When not provided in the original paper, the standard deviation was estimated from the reported standard error (Column 3) using the number of users and trials involved in the experiments as reported in Table 2-Column 2.

|       | Time Delay (ms) | Standard Error | Reference |
|-------|-----------------|----------------|-----------|
| GY    | 40 ± 371        |                | [26]      |
|       | 326 ± 237       |                | [26]      |
| HY    | 180 ± 164       | 30             | [23]      |
|       | 530 ± 164       | 30             | [23]      |
|       | 360 ± 100       | 10             | [24]      |
|       | 530 ± 250       | 25             | [25]      |
|       | −734 ± 1518     | 160            | [28]      |
|       | 50 ± 317        |                | [26]      |
|       | 349 ± 210       |                | [26]      |
| TY    | 470 ± 164       | 30             | [23]      |
|       | 700 ± 164       | 30             | [23]      |
|       | 420 ± 100       | 10             | [24]      |
|       | 610 ± 200       | 20             | [25]      |
|       | −571 ± 1518     | 160            | [28]      |
| TR    | 280 ± 164       | 30             | [23]      |
|       | 390 ± 164       | 30             | [23]      |
|       | 500 ± 300       | 30             | [24]      |
|       | −177 ± 1518     | 160            | [28]      |
| MLC   | 610 ± 380       | 38             | [24]      |
|       | 700 ± 200       | 20             | [25]      |
|       | −447 ± 1138     | 120            | [28]      |

References [12,16,21,22,30] did not provide precise quantitative results. Relevant findings from these works will be qualitatively considered in the discussion when appropriate.

Other authors reported anticipation times directly between variables and the trajectory heading. In [29], authors reported an anticipation of 202 ± 83 ms from Gaze Yaw to Head Yaw, 444 ± 202 ms from Gaze Yaw to Trunk Yaw and 212 ± 162 ms from Head Yaw to Trunk Yaw. In [19], Head Yaw was found to anticipate Trajectory Heading by 207 ± 165 ms. In [20], Gaze Yaw and Head Yaw were found to anticipate Trajectory Heading by 557 ± 160 ms and 264 ± 134 ms respectively.

Regarding the IMU-based studies, an anticipation of head, upper back and lower back orientation onsets to the actual turning time of 204 ms, −69 ms and 5 ms were reported in [10]. Anticipation of head turning with respect to pelvis turning was found to be of 528 ± 385 ms in [32].

With respect to geographical anticipation, [17] reported a time anticipation of Gaze Yaw, Head Yaw and Trunk Yaw onsets to the corner pivot of 500 ± 300 ms, 800 ± 200 ms and 1500 ± 200 ms, respectively. From them, we estimated the corresponding anticipation in distance at 0.35 ± 0.21 m, 0.57 ± 0.14 m and 1.07 ± 0.14 m. [27] reported an anticipation distance of Head Yaw onset to the corner pivot of 0.3 ± 0.3 m. Other values regarding the anticipation to the corner pivot have been reported in [30,31], however, they lacked precision and will be qualitatively considered in the discussion section when appropriate.

## 3. Results

Table 4 contains data about the time synchronization of upper body kinematical variables (Gaze Yaw, Head Yaw, Trunk Yaw, Trunk Roll). Additionally, we included available data about the

anticipation time of each variable with respect to the estimated body Trajectory Heading (TJH) and the Medio-Lateral COM displacement (MLC). Values in bold were directly reported in the corresponding reference. The others were calculated by combining results from Table 3 as described in Section 2.5.

**Table 4.** Time anticipation (column 3, mean $\pm$ standard deviation) between gaze (GY), body segments (TY, TR), trajectory heading (TJH) and body center of mass medio-lateral displacement (MLC). Values in bold were directly reported in the corresponding reference. The others were calculated by combining results from Table 3. Each value is accompanied by the corresponding reference into brackets.

| Time (ms) | HY | TY | TR | MLC | TJH |
|---|---|---|---|---|---|
| GY | $10 \pm 195$ [26]<br>$23 \pm 125$ [26]<br>**$202 \pm 83$** [29]<br>$293 \pm 84$ [20] | **$444 \pm 202$** [29] | | | **$557 \pm 160$** [20] |
| HY | | $290 \pm 90$ [23]<br>$170 \pm 90$ [23]<br>$60 \pm 55$ [25]<br>$80 \pm 132$ [25]<br>$163 \pm 831$ [28]<br>**$212 \pm 162$** [29] | $100 \pm 90$ [23]<br>$-140 \pm 90$ [23]<br>$140 \pm 221$ [24]<br>$557 \pm 831$ [28] | $250 \pm 300$ [24]<br>$170 \pm 132$ [25]<br>$287 \pm 814$ [28] | **$207 \pm 165$** [19]<br>**$264 \pm 134$** [20] |
| TY | | | $-190 \pm 90$ [23]<br>$-310 \pm 90$ [23]<br>$80 \pm 221$ [24]<br>$394 \pm 831$ [28] | $190 \pm 300$ [24]<br>$90 \pm 110$ [25]<br>$124 \pm 814$ [28] | |
| TR | | | | $110 \pm 201$ [24]<br>$-270 \pm 814$ [28] | |

An aggregated analysis reveals that Gaze Yaw significantly anticipates Head Yaw by 179 ms (*p*-value < 0.01). The 95% confidence interval for the anticipation is [54 ms, 305 ms]. However, the shortest value ($10 \pm 195$ ms) reported in [26] could be explained by the nature of sudden turn decisions which just made subjects re-orient segments faster, as they were told the direction to turn using a visual cue that was emitted at the turning point. Therefore, if we consider only the remaining experiments, where subjects were informed at the beginning of the experiment about the turn to take, and hence, had enough time in advance to schedule the turning maneuver, the combined analysis reveals that Gaze Yaw significantly anticipates Head Yaw by 198 ms (*p*-value < 0.01, 95% confidence interval [68 ms, 327 ms]).

Head Yaw significantly anticipated Trunk Yaw by 149 ms (*p*-value < 0.01, 95% confidence interval [47 ms, 251 ms]). As before, some results ($170 \pm 90$ ms, $60 \pm 55$ ms, $80 \pm 132$ ms, reported in [23–25]) were obtained from experiments where users had to turn immediately after a signal/auditive clue. Excluding these results, Head Yaw significantly anticipated Trunk Yaw by 271 ms (*p*-value < 0.001, 95% confidence interval [117 ms, 424 ms]). On the other side, Head Yaw significantly anticipated Trajectory Heading by 241 ms (*p*-value < 0.05, 95% confidence interval [37 ms, 445 ms]). Conversely, Head Yaw was not significantly different from Trunk Roll (mean anticipation 11 ms, 95% confidence interval [−178 ms, 200 ms]), nor from Medio-Lateral COM Displacement (mean anticipation 185 ms, 95% confidence interval [−49 ms, 420 ms]).

Trunk Yaw did not significantly anticipate Medio-Lateral COM Displacement (mean anticipation 102 ms, 95% confidence interval [−98 ms, 302 ms]).

Finally, Trunk Roll significantly anticipated Trunk Yaw by 216 ms (*p*-value < 0.01, 95% confidence interval [84 ms, 348 ms]). Trunk Roll did not significantly anticipate Medio-Lateral COM Displacement (mean anticipation 88 ms, 95% confidence interval [−295 ms, 471 ms]).

In relation to the geographical anticipation, reported values were sparse and lacked precision in most references. For that reason, we decided not to aggregate them, deferring their consideration for the discussion in a qualitative sense.

## 4. Discussion

From the analysis of the available data we can state the following.

### 4.1. Gaze Yaw Is the Earliest Predictor of Walking Turns; However, Existing Evidence Does Not Allow to Quantify the Anticipation and Its Reliability

Our analysis revealed that Gaze Yaw anticipated Head Yaw up to 179 ms in aggregated terms. However, we have found that in one of the considered studies [26] both events were almost simultaneous. Additionally, eye movements have been found to depend greatly on ambient factors that can even reverse the usual sequence of anticipatory movements as reported in [17], where GY happens even after TY. This behavior may be explained as gaze does not develop standalone in turn negotiation, being intrinsically coordinated with the head progression [36,37]. The collected results also show that Gaze Yaw anticipates Trunk Yaw by 444 ms [29] and Trajectory Heading by 557 ms [20]. However, only a single study is available for each of these values, and thus, no generalization can be made from them. Therefore, Gaze Yaw anticipation to walking turns needs further research to be confirmed, perhaps oriented to the common consideration and quantification together with head turning.

### 4.2. Trunk Yaw Is Not Valid to Predict Walking Turns as It Tracks the Actual Trajectory of the Subject

In our study we did not find quantitative data regarding the synchronization between Trunk Yaw and Trajectory Heading. In any case, considering the analyzed literature, we can consider that both evolve synchronously, and Trunk Yaw is thus not valid to predict walking turns.

Papers [19,20] reported no significant difference between Trunk Yaw and Trajectory Heading. This was additionally supported in [29], where the authors reported a short delay ($-26.52 \pm 73.45$ ms) between Trunk Yaw and pelvis rotation trajectory that was not significantly different from zero. Results in [10] were also consistent with this statement, showing that the turn can be detected from the upper trunk with a delay of 69ms and from the lower trunk with an anticipation of 5ms.

An apparently contradictory result was reported in [17,31], where Trunk Yaw was found to predict the geographical corner pivot from 0.9 m to 1.14 m. This result can lead to think that Trunk Yaw anticipates walking turns. However, they may be explained by different turn negotiation strategies or simply by the experimental conditions. For instance, in some experiments described in [31], subjects had to walk along a trajectory painted on the floor that actually started to bend in the turning direction before the corner pivot, using concentric circles (0.5 m of radius) around it. Therefore, users had to modify the walking trajectory before the corner pivot, which could justify the anticipated detection Trunk Yaw onset.

Our analysis also showed that Trunk Yaw onset was not significantly different from Medio-Lateral COM Displacement onset, further confirming the statement, as Medio-Lateral COM Displacement is also an indicator of the change in the trajectory of the subject.

### 4.3. Head Yaw Is a Reliable Choice to Consistently Anticipate Heading Direction in Real Time by Around 200 ms

It is well known in neuroscience that the head anticipates motion direction to provide a stable reference frame that helps to coordinate the motion of the other body segments (see for instance [12,28]). As expected, results from the addressed studies are consistent with this behavior: changes in Head Yaw precede changes in other upper body segments and in the trajectory orientation. This was even found under different experimental conditions (eyes open/closed or walking forwards/backwards [22]). Discrepancies to this general finding have been occasionally reported. However, they can be attributed to the particularities of the addressed experiments. In [17], Trunk Yaw was delayed 600 ms from Head Yaw. However, in this study, the experiments were designed to study the effect on locomotor coordination when pushing a wheelchair and this may perhaps have biased the results. In [16], Head Yaw was found to be synchronized with Trunk Yaw, however, in this work, subjects were involved in a very particular steering maneuver (obstacle avoiding).

Initial studies found an anticipation of about 440 ms from Head Yaw to Trunk Yaw [21]. However, results of [23,28,29] advance that this anticipation is expected to be lower. An interesting remark is that results of [29] agree with those of [23,28], even though they were obtained under a completely different experimental setup, involving a greater diversity of turns.

Our aggregated analysis revealed a significant anticipation of Head Yaw to Trunk Yaw of 149ms, on average. This anticipation increases to 271 ms if we consider only preplanned turns.

The results of these studies were also consistent with others obtained from different experimental setups using body-worn inertial sensors [10], where head onset anticipated the upper back yaw onset by 273 ms. A recent work [32], also based on inertial sensors, reported a greater anticipation (528 ms on average from the head yaw to the pelvis yaw). However, the experimental setup is dramatically different, including a different definition to detect onsets of variables and advanced machine learning methods for the data analysis, and cannot be easily compared.

The aggregated analysis reveals also that Head Yaw significantly anticipates Trajectory Heading by 241ms on average. This anticipation is consistent with the Head Yaw-Trunk Yaw anticipation, thus providing an additional support about the actual evolution of Trunk Yaw with the Trajectory Heading. However, a noteworthy result is that the aggregated analysis showed that Head Yaw does not significantly anticipate Medio-Lateral COM Displacement. Medio-Lateral COM Displacement estimation is in itself a complex problem that may be the reason for this contradictory result.

### 4.4. Trunk Roll Anticipates Trajectory Heading Similarly to Head Yaw but with Less Reliability

We have found conflicting results about the role of the Trunk Roll during walking turns. In [23], Trunk Roll anticipated Trunk Yaw, however, in [24] the sequence was reverted. Results from [28] showed that Trunk Roll is the last change registered in the analyzed biomechanical chain, even after Medio-Lateral COM Displacement. In [16], Trunk Roll was found to be negligible, although in this paper anticipation was monitored in a special steering case (circumventing an obstacle). These could indicate that the role of Trunk Roll in the turning strategy may vary depending on the actual conditions of the turn. In any case, our combined analysis showed that Trunk Roll anticipation is not significantly different from Head Yaw anticipation. Thus, Trunk Roll can be considered an additional predictor of walking turns with a similar anticipation to that obtained from Head Yaw, but with less reliability.

### 4.5. Head Yaw Predicts the Geographical Turning Point at a Constant but Unknown Distance

The paper [38] can be considered a pioneering study about geographical anticipation. In this case, anticipation was measured against the point defined by the intersection of the rectilinear part of the actual path that the subject described before and after the turn. Results show that Head Yaw started later and later as speed increased, from 3.8 s at 1.4 m/s to 0.44 s at 9.7 m/s. However, when it was plotted versus the distance from the corner, the waveforms aligned almost perfectly; thus, being the first work to support that the anticipation of Head Yaw to the turning point is constant in distance, not in time. Numerical results showed an anticipation of between 2 m and 9 m depending on the individual subject. However, these experiments were designed using a virtual reality environment that sometimes supposed very high velocities and were thus not representative of normal walking conditions.

Later studies supported what has been called the "spatial invariance" hypothesis, which means that Head Yaw starts to vary at a constant distance to the turning point. However, reported anticipation distances did not allow to quantify it with precision. Head Yaw anticipation to the corner pivot was quantified in 0.3 m [27] and 1.1 m [31]. This anticipation was found to be invariant to the walking velocity [27] and to the turning angle (from 22° to 135°) [31]. Interestingly, this did not depend on the turn strategy, as there were no significant differences in results from experiments where users were told to walk along a trajectory painted on the floor from results from other experiments where users could freely negotiate the turn [31]. The work [17] reported a larger anticipation distance of 2.1 m. Similarly, in [30], authors reported anticipation to the turning point in the range of 2–4 m. These results,

consistent for each work but greatly diverging between different references, might advance that the actual anticipation distance may depend on the type of turn and its environmental constraints.

## 5. Considerations about Turn Prediction Systems from Upper Body Kinematics in Human-Robot Interaction

The previous analysis finds a direct application in human-robot interaction. It revealed that Head Yaw is the earliest variable affected by the turning maneuver with an adequate degree of reliability. As Head Yaw predicts anticipated trajectory heading, based in the correlation-based studies in Table 1, it can be used to elaborate advanced predictive models of the spatial occupancy of the subject for the "goal-free motion prediction" problem [3]. At the moment, quite simplistic models have been used for this purpose, mainly based on the projection of the current position of the subject using the current velocity and trajectory heading direction [1,3]. Head Yaw/Trunk Roll can also be used to predict in time an upcoming turning maneuver (onset-based studies, Table 1), and therefore, can be used to predict the specific destination or walking direction when a finite number of possibilities are available ("goal-based motion prediction" problem [3]).

From a safety point of view, robot motion planning in human robot collaboration that considers turn prediction from Head Yaw or Trunk Roll has to be used with caution. Head Yaw is the preferred behavior during turning [19]. However, head rotations are neither a necessary nor a sufficient component of steering control [39]. With regard to Trunk Roll, the reliability of the anticipation is lower, even though the anticipated detection of walking turns can be useful to increase comfort or perceived safety. An appropriate sensor-based dynamic robot motion planning [40] with improved proxemic behavior can be achieved by controlling the movement synchronization or the separation distance with the human. For a walking velocity of 1.6 m/s, an anticipation to the trajectory change around 200 ms may permit to forecast the position of the subject about 0.32 m ahead. This anticipation can be relevant for human-robot teams interacting at an intimate distance (up to 0.46 m), but also at a personal distance (0.46 m to 1.2 m) or even at a social distance (1.2 m to 2.4 m) [41]. However, for larger distances of separation between the human and the robot, turn anticipation from these variables is not expected to suppose a relevant improvement.

Many technical alternatives are available for the estimation of Head Yaw and Trunk Roll in human-robot interaction applications. Optical motion capture systems require a precise deployment of cameras and complex calibration procedures, which can be a limitation in some unstructured interaction scenarios. As an alternative, wearable sensors attached to the body in clothes, hats, headbands, glasses, etc., have been successfully applied to the early detection of turns [10,32]. The downside is that inertial sensors suffer from signal drift issues. However, this problem is diminishing due to technological advances in the sensors and in the processing methods [42].

## 6. Scope of the Study

We have considered only healthy adult subjects (with an average age of about 27 from Table 2, column 2). Turning strategies depend on the grade of maturity of children [21] and may also vary for the elderly [43,44]. In any case, some results showed that anticipation may not be so different for the elderly (Head Yaw-Trunk Yaw delay was found similar for young and elderly people [28]). Additionally, our analysis was limited to subjects walking at normal velocities (average walking speed of around 1.2 m/s from Table 2, column 2). No studies were found addressing this variable under significant different walking velocities to permit to extrapolate this result to a general case.

For our study, we have combined results from experiments addressing diverse trajectories: corners at different angles, 180° turns, limaçon, free trajectories, etc. (see Table 2, column 3). Thus, our analysis can be interpreted from the point of view of walking performance under general turning conditions. Likewise, all the experiments were run in a laboratory setup. It is open to investigation if they hold for daily living conditions, although [11] supported this after a preliminary study in an industrial environment. Another relevant question is that the experiments were designed such that subjects

had to undertake turning maneuvers in previously unknown environments and conditions. Re-test experiments to analyze the effect of habits or custom turns over the anticipation were not found in the literature. However, we hypothesize that gaze/head anticipation would still remain in custom turns, similar to the situation where gaze/head anticipation still happens in turns without vision ([20,23]).

The kinematics of head and trunk have been estimated in the addressed studies from optical motion capture systems using different sets of body markers and anatomical models (see Table 1). For our study, we have assumed that all these variables have been properly estimated in the original work. However, the Body Centre Of Mass estimation from body kinematics is in itself a complex problem. In the analyzed studies, different methods were used for that purpose to estimate the medio-lateral displacement of body center of mass (see Table 1): from a 14-segment anthropometric model including legs and arms in [24], from four trunk markers in [25], from head, neck and trunk segment markers in [16] and from a "two-segments plus foot" anthropometric model [28]. Likewise, no clear consensus has been found about how to define the actual trajectory of the subject or the actual time of trajectory change: from the time when the subject is at the maximum curvature point of the actual trajectory [21,22], using the tangent of head trajectory [12], from "the displacement vector of the midpoint between the three trunk markers" [19], from the pelvis direction [20] or from "the projection of the middle point of the head on the floor" [29]. It is not clear how these different procedures to estimate body center of mass or trajectory heading may have affected the results. Finally, the methods used for the estimation of the kinematics of the gaze (oculography and eye tracking) were quite diverse and difficult to compare.

Different signal processing procedures have been applied in the addressed works to analyze the time synchronization of kinematical variables (Turn Onset, Cross Correlation). What is more, in some works, turn onsets were timed from a different reference event. This makes a raw comparison of the quantitative reported values difficult. However, in our study, we differentially compared the turn onset times found in each study. This makes it irrelevant whether onsets were defined from different events, or even when different techniques were used to define them. Regarding the geographical anticipation analysis, the considered studies addressed the initiation of turning with respect to a geographical landmark (usually a corner pivot) defined consistently throughout the different works.

The previous factors, and other issues detected in our study such as missing data or varied experimental conditions, make the quantitative aggregated results eventually provided in the discussion only valid as indicative values. Additionally, standard deviation values reported in Table 4 have been occasionally calculated using a correlation factor of 0.85. This implies that variables (HY, TY, TR, MLC) are supposed to be highly correlated.

## 7. Conclusions

Human-Robot interaction, and perhaps other scenarios where humans and machines interact, may greatly benefit from automatic methods for human movement prediction. Even though human movement performance has been extensively addressed in the neuroscientific and biomedical fields, it has rarely been considered in this context and there is presently a real need to reinterpret the state-of-the-art from the point of view of obtaining practical conclusions.

In this paper, we particularly addressed the control mechanisms of the human body that occur during walking turns. To correctly negotiate changes of direction, some adaptations develop in the upper body prior to the turning point. These adaptations are reflected in the kinematics of those upper body segments, and thus, can be detected in real time to predict the occurrence of the turn. However, some questions were still unanswered: what variables can be reliably used for that purpose and how long in advance can one expect to anticipate the turn from them? Moreover, what is the expected impact for human-robot interaction of anticipating walking turns from upper body kinematics?

To contribute to these research questions, we have compiled, analyzed and discussed a great deal of evidence about the quantitative anticipation of walking turns from upper body kinematics. On the one hand, our study supports the feasibility to anticipate the trajectory direction in unrestricted

walking conditions from Head Yaw and Trunk Roll kinematics by around 200 ms in time. On the other, it supports that they are the only kinematical major variables in the upper trunk that can be used for that purpose from existing evidence thus far, as pelvis and trunk yaw have been found to evolve simultaneously with the trajectory of the subject. Regarding Gaze Yaw, existing studies advanced that it is perhaps the earliest predictor; however, more work is needed to quantify its utility for practical applications.

Anticipating turns from Head Yaw and Trunk roll is a plausible, feasible and promising approach using state-of-the-art sensors and simple signal processing methods. Its main contributions are expected to occur in close proximity interaction scenarios, helping the robot to exhibit appropriate proxemic behavior. However, this approach has to be used with caution from the point of view of human safety, as anticipated head rotation is not a necessary part of steering control and trunk roll does not exhibit a clear consistent behavior in the considered research.

**Supplementary Materials:** https://gitlab.com/simur/applsci-419915/blob/master/applsci-419915.zip.

**Author Contributions:** The authors contributed equally to this work.

**Funding:** This research received no external funding.

**Conflicts of Interest:** The authors declare no conflict of interest.

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
