# Peer review of "Walking Turn Prediction from Upper Body Kinematics: A Systematic Review with Implications for Human-Robot Interaction"

_applsci, doi:10.3390/app9030361_

Round 1
Reviewer 1 Report
This paper reports a survey to address the control mechanisms of the human body during walking turns. It found that Gaze Yaw and Head Yaw are predictors of walking turns, but Trunk Yaw is not.
Overall the paper is easy to read and some new findings have been revealed as a result of the survey.
Author Response
This paper reports a survey to address the control mechanisms of the human body during walking turns. It found that Gaze Yaw and Head Yaw are predictors of walking turns, but Trunk Yaw is not.
Overall the paper is easy to read and some new findings have been revealed as a result of the survey.
Dear Reviewer,
Thank you so much for your thorough revision and your comments. Certainly these have been our goals with this study.
Reviewer 2 Report
This paper presents a review of the state of the art on methods for walking turn prediction, based on upper body kinematics. The authors pursue a systematic method for the search, filtering and selection of relevant papers. This is followed by presenting a summary of the quantitative and qualitative results extracted and extrapolated from the final selected papers, in a comparable manner.
The topic is of importance to the field, and it is indeed of use to have the state-of-the-art reviewed in detail, their results aggregated, and cumulative conclusions made. The paper is generally well written and structured. Methods are sound and presented in detail, discussion is thorough, with results and conclusions well presented and summarised. There are, however, some issues present that need to be resolved:
Major points:
- Implications for Human-Robot Interaction are very minimally discussed. HRI is a very wide area, including many different types of robots and applications. It is unclear how the results of this paper would translate into any of these scenarios. What is discussed in section 5 with regards to HRI, does not depend on the results obtained. Most of it are obvious considerations within HRI. The best case to include the HRI aspect would’ve been additional experiments or results pertaining specifically to walking turn prediction in an HRI scenario. In absence of that, I believe the authors should either remove the HRI aspect, or extend the discussion with further details.
- Further transparency on how the paper search was performed, and how the results were filtered would be very helpful to readers. Examples follow:
1. “Three additional papers were added after an Internet search” is a very vague sentence; where was the search performed? It is rather odd that other sources on the internet only added 3 papers to the list. More details will help.
2. The exclusion based on abstracts is vague; what was the criteria here? Was this done by one person or multiple people? Was this done systematically, i.e. multiple people reviewing papers, so that more than one person is judging each paper’s abstract – or was it all done by the same person and without double checks? How did you ensure that important details are not missing from the abstract?
3. PRISMA is only mentioned in the flow diagram figure caption (which incidentally has the wrong figure number). It should be described briefly within the text and cited.
4. Going from 400 down to 27 is a very narrow filter. To better understand this, best practice would be to make the entire list of papers inspected (all 400) available. This can be a simple txt file or a spreadsheet with the citations, to be uploaded as an appendix document, or kept on your own servers and linked to from the paper document.
5. Would be useful to specify which reference number ended up in which category.
These are just examples. A proper revision is recommended.
- The authors briefly mention that these results are from able-bodied, healthy participants and that, obviously, results would be different in other cases. However, prosthetics and orthotics are mentioned as areas where these results could be used. These two statements contradict. Anticipatory movements of the upper body are expected to be very different in those with movement disabilities.
- It would be worthy to discuss the effects of learning and habits. The results presented are from experiments where participants are facing a new task in a new environment. Can we expect the anticipatory movements to be different, or even non-present when a user is in a familiar, repetitive environment? This could particularly apply to Gaze Yaw and Head Yaw, as those are mainly for visual inspection.
Minor points:
- Would be very useful to have the aggregated data, and calculated missing data made available with the paper.
- A general revision of the paper is recommended, there are a few miswritten words and some confusing sentences. Furthermore, the writing style seems to change slightly in different sections – might be due to work by multiple authors – a revision will help.
Author Response
This paper presents a review of the state of the art on methods for walking turn prediction, based on upper body kinematics. The authors pursue a systematic method for the search, filtering and selection of relevant papers. This is followed by presenting a summary of the quantitative and qualitative results extracted and extrapolated from the final selected papers, in a comparable manner.
The topic is of importance to the field, and it is indeed of use to have the state-of-the-art reviewed in detail, their results aggregated, and cumulative conclusions made. The paper is generally well written and structured. Methods are sound and presented in detail, discussion is thorough, with results and conclusions well presented and summarised.
Thank you so much for your enriching and thorough revision.
Certainly, these have been indeed the main objectives of our work. You point us some interesting aspects that may contribute to improve the study. We have addressed them as detailed below.
There are, however, some issues present that need to be resolved:
Major points:
- Implications for Human-Robot Interaction are very minimally discussed. HRI is a very wide area, including many different types of robots and applications. It is unclear how the results of this paper would translate into any of these scenarios. What is discussed in section 5 with regards to HRI, does not depend on the results obtained. Most of it are obvious considerations within HRI. The best case to include the HRI aspect would’ve been additional experiments or results pertaining specifically to walking turn prediction in an HRI scenario. In absence of that, I believe the authors should either remove the HRI aspect, or extend the discussion with further details.
Certeinly, perhaps this part of the discussion was not correctly focused in the original draft. Our purpose with this paper is to settle some basic practical foundations that may help future experimentation and eventually worthy advances in the field of HRI.
We are convinced that the study has direct implications for HRI. First at all, turn anticipation has to be addressed in this framework from head yaw/trunk roll. Second, head-yaw can be considered from two perspectives: as a predictor of the displacement heading or (complemented with trunk roll) as an indicator of a future turn. Third, from the point of view of safety these kind of prediction may present some reliability limitations. However, the approach can be useful to improve the perceived safetey (proxemics) as it has been shown before. Fourth, in open environments upper body kinematics sampling is becoming feasible (wearable sensors).
We have tried to focus better this discussion around these facts, omitting obvious considerations and concentrating on the most relevant implications. The section has been completely rewritten (lines 423-452).
- Further transparency on how the paper search was performed, and how the results were filtered would be very helpful to readers. Examples follow:
1. “Three additional papers were added after an Internet search” is a very vague sentence; where was the search performed? It is rather odd that other sources on the internet only added 3 papers to the list. More details will help.
2. The exclusion based on abstracts is vague; what was the criteria here? Was this done by one person or multiple people? Was this done systematically, i.e. multiple people reviewing papers, so that more than one person is judging each paper’s abstract – or was it all done by the same person and without double checks? How did you ensure that important details are not missing from the abstract?
3. PRISMA is only mentioned in the flow diagram figure caption (which incidentally has the wrong figure number). It should be described briefly within the text and cited.
4. Going from 400 down to 27 is a very narrow filter. To better understand this, best practice would be to make the entire list of papers inspected (all 400) available. This can be a simple txt file or a spreadsheet with the citations, to be uploaded as an appendix document, or kept on your own servers and linked to from the paper document.
5. Would be useful to specify which reference number ended up in which category.
These are just examples. A proper revision is recommended.
Thak you so much for highlighting this shortcoming. We regret the poor description of the methodology we have made and all the missing aspects in the original draft . We have rewritten the section including the relevant datails, specifically considering the mentioned points (lines 114, 117-130). Also,following your advice, we have also ellaborated a file containing the 400 considered references, that has been made publicly available in GITLAB (the link is included in the "Supplementary Materials" section, line 656).
- The authors briefly mention that these results are from able-bodied, healthy participants and that, obviously, results would be different in other cases. However, prosthetics and orthotics are mentioned as areas where these results could be used. These two statements contradict. Anticipatory movements of the upper body are expected to be very different in those with movement disabilities.
We tried to mention prosthetics and orthotics only as potential application fields. Of course, pathological or disabled gait greatly diverges from normal gait, and results from this study are not, with almost certain security, of direct application to this new scenario. Considering that the possible application to this field is mentioned in the introduction (lines 46-48), we have decided to remove this paragraph.
- It would be worthy to discuss the effects of learning and habits. The results presented are from experiments where participants are facing a new task in a new environment. Can we expect the anticipatory movements to be different, or even non-present when a user is in a familiar, repetitive environment? This could particularly apply to Gaze Yaw and Head Yaw, as those are mainly for visual inspection.
This aspect has not been considered in the literature and, to our kowledge, has rarely been considered in gait analysis. A new study should be designed aroud this idea to have a definitive answer. Anyhow, we hypothesise that gaze/head anticipation would still remain (perhaps in other anticipation ranges), similar to the case where anticipation is still present in turns during experiments performed without vision.
We have considered this interesting idea at line 446-471.
Minor points:
- Would be very useful to have the aggregated data, and calculated missing data made available with the paper.
We have made publicly available in GITLAB a zip file containing:
A pdf file with a dump of the execution of the script used for the aggregatted analysis from the data included in Table 4.
An excel file that contains the values included in table 3 and equations to calculate from them the values reported in Table 4.
The list of references (pdf) considered for the study.
The link is included in the "Supplementary Materials" section, line 656.
Additional comments:
During the revision we have realized that the data reported in the original draft in Table 4 was truncated as a consequence of transcription errors. It had no implications over the aggregatted reported values, as they were calculated from the right values. We have corrected truncated values in Table 4 in the present draft.
During the final revision of the draft, we also found a "bug" in the calculations of one value of Table 3 and one value of Table 4. For the sake of correctness, we have included in the revised draft the right values, and we have considered the new values in the calculations of the aggregated values. This corrections present no actual implications over the findings of the paper, except for slight changes in some numerical values.
All the changes have been highlighted in the new draft.